# A Systematic Investigation Unveils High Coinfection Status of Porcine Parvovirus Types 1 through 7 in China from 2016 to 2020

Jixiang Li,[a] Yanzhao Xiao,[a] Ming Qiu,[a] Xinshuai Li,[a] Shubin Li,[a] Hong Lin,[a] Xiangdong Li,[a,b,c,d] Jianzhong Zhu,[a,b,c,d] Nanhua Chen[a,b,c,d]

[a]College of Veterinary Medicine, Yangzhou University, Yangzhou, Jiangsu, China
[b]Joint International Research Laboratory of Agriculture and Agri-Product Safety, Yangzhou, Jiangsu, China
[c]Comparative Medicine Research Institute, Yangzhou University, Yangzhou, Jiangsu, China
[d]Jiangsu Co-innovation Center for Prevention and Control of Important Animal Infectious Diseases and Zoonoses, Yangzhou, Jiangsu, China

Jixiang Li and Yanzhao Xiao contributed equally to this article. The author order was determined by their equal but gradated contributions for this paper.

**ABSTRACT** Porcine parvovirus genotype 1 (PPV1) causes reproductive disorder in swine and is prevalent in China. Recently, six new genotypes of PPVs (PPV2 through PPV7) have also been detected in Chinese swine herds. However, the coinfection status of all these seven genotypes of PPVs (PPV1-7) in China was not clarified yet. In this study, we developed a panel of PPV1–7 PCR assays with satisfied specificity, sensitivity and reproducibility and then applied to the detection of PPV1–7 in 435 clinical samples collected from eight provinces of China in 2016–2020. A total of 55.40% samples (241 out of 435) were PPV positive, while PPV2 and PPV3 (both 22.53%) belonging to the genus of *Tetraparvovirus* were the most prevalent genotypes. Noticeably, PPV1–7 strains were more prevalent in nursery and finishing pigs than in suckling pigs. In addition, coinfection could be detected in all eight provinces and 27.36% (119/435) samples were coinfected with two to five genotypes of PPVs. Meanwhile, the coinfection of PPVs with PCV2 was 22.30% (97/435). Twenty complete genomes of representative PPV1–7 were determined, and phylogenetic analysis confirmed the genotyping results by sequence comparisons and PCR assays. Remarkably, the PPV7 HBTZ20180519-152 strain from domestic pig was recombined from parental JX15-like and JX38-like isolates from wild boars. Selective pressure analysis based on VP2 sequences of PPV1–7 showed that they were predominantly under negative selection, while few positive selection sites could be detected in VP2 of PPV7. Overall, this systematic investigation unveils high prevalence and coinfection of PPV1–7 in China from 2016 to 2020.

**IMPORTANCE** Porcine parvoviruses (PPVs) are prevalent in China associating with reproductive failure in swine. The coinfection of seven genotypes of PPVs (PPV1-7) might have synergistic effects on PPV1 associated SMEDI syndrome. However, the coinfection status of PPV1–7 in China is not clear yet. This study showed that PPV1–7 strains are highly prevalent (55.40%) in China and mainly in nursery and finishing pigs in recent years. In addition, the coinfections of different genotypes of PPVs (27.36%) and PPVs with PCV2 (22.30%) are common. Geographic analysis indicated that different genotypes of PPVs are widely cocirculating in China. Intriguingly, a PPV7 strain from the domestic pig was detected as a recombinant from two wild boar isolates. Selective pressure analyses showed that PPV1–7 are mainly under purifying selection. Our findings provide the first systematic investigation on the prevalence, coinfection, and evolution of PPV1 through PPV7 in Chinese swineherds from 2016 to 2020.

Address correspondence to Nanhua Chen, chnhlh@126.com, Xiangdong Li, xiaonanzhong@163.com, or Jianzhong Zhu, jzzhu@yzu.edu.cn.

**KEYWORDS** porcine parvovirus genotypes1–7, prevalence, coinfection, genome, recombination, selective pressure

*P*aroviridae is a family of small nonenveloped DNA viruses with a single-stranded linear genome. According to virus classification and taxonomic proposals approved by the International Committee on Taxonomy of Viruses (ICTV) Executive Committee in July 2019 (1), the *Parvoviridae* family comprises three subfamilies: *Parvovirinae*, *Hamaparvovirinae* and *Densovirinae* (https://talk.ictvonline.org/taxonomy/). The *Parvovirinae* and *Hamaparvovirinae* subfamilies contain viruses infecting vertebrates, while the *Densovirinae* subfamily contains viruses infecting arthropods. Viruses that infect pigs are categorized within the *Protoparvovirus*, *Tetraparvovirus*, and *Copiparvovirus* genera of *Parvovirinae* subfamily and the *Chaphamaparvovirus* genus of *Hamaparvovirinae* subfamily.

The genomes of porcine parvoviruses (PPVs) are approximately 4.0–6.3 kb in size, which contain two major open reading frames (ORFs). ORF1 encodes nonstructural proteins NS1, NS2 and NS3, while ORF2 encodes structural proteins VP1 and VP2 (2). PPV genotype 1 (PPV1) belongs to the genus of *Protoparvovirus* (species of *Ungulate protoparvovirus 1*), which was first isolated as a cell culture contaminant in Germany in 1965 (3). PPV1 is the major etiological agent of SMEDI syndrome (stillbirths, mummification, embryonic death and infertility). Before the 21st century, PPV1 was the sole representative of *Parvovirinae* members. In the last 2 decades, another six novel PPV genotypes (PPV2 to PPV7) have been successively identified by the sequence-independent PCR and high-throughput sequencing methods (4, 5).

In 2001, PPV2 was accidentally PCR-amplified in swine sera from Myanmar (6). In 2008, PPV3 (porcine hokovirus [PHoV]) was first discovered in Hong Kong (7). Both PPV2 and PPV3 belong to the genus of *Tetraparvovirus*. PPV4 and PPV5 were reported in 2010 and 2013 in United States (8, 9), and PPV6 was initially identified from aborted pig fetuses in 2014 in China (10). The closely related PPV4, PPV5 and PPV6 were all clustered within the genus of *Copiparvovirus*. PPV7 was first detected in healthy adult pigs in 2016 in United States (11), which was recently grouped into the genus of *Chaphamaparvovirus* in the subfamily of *Hamaparvovirinae*. Unlike PPV1 which is a main cause responsible for reproductive failures in pigs, the knowledge regarding the pathogenicity of novel PPVs (PPV2 to PPV7) is limited given that virus isolations and experimental infections have not been executed (5).

In China, PPV1 was first identified in 1983 (12). Since then, it has been widely spread in Chinese swine herds. Furthermore, all the newly recognized PPVs have also been detected in China (10, 12–15). Different PPV prevalence becomes a serious threat in China. Therefore, a systematic surveillance of PPV prevalence is required to clarify the real-time epidemiology of PPV in Chinese swine herds, which is also beneficial for setting up feasible and effective control strategies. In this study, we developed a panel of PCR assays for the differential detection of PPV1 through PPV7. Clinical samples collected from different ages of pigs in eight provinces of China from 2016 to 2020 were evaluated by PCR methods and confirmed by sequencing. The prevalence and coinfection status of PPV1–7 in Chinese swine herds were determined. Furthermore, a total of 20 nearly complete genomes of PPV1–7 were sequenced and submitted to multiple alignment, phylogenetic analysis, recombination test and selective pressure evaluation.

## RESULTS

**Development of the panel of PPV1–7 PCR assays.** Corresponding fragments for PPV1 through PPV7 could be specifically amplified by the new PPV1–7 PCR assays (Fig. S1) using primers shown in Table S1. Specificity evaluation showed that the new assays could detect and differentiate PPV1 through PPV7, but no specific fragments could be detected when other common porcine viruses (PCV, PRV, PEDV, CSFV, and PRRSV) were tested (Fig. S2). By using 10-fold serial dilutions of the positive-control, the detection limits of the panel of PCR assays for all genotypes of PPVs were ranged

**TABLE 1** Prevalence of PPV1–7 in tissue samples collected from 2016 to 2020

| Yrs | PPV1 | PPV2 | PPV3 | PPV4 | PPV5 | PPV6 | PPV7 | Total |
|---|---|---|---|---|---|---|---|---|
| 2016 | 1/15 (6.67%) | 0/15 (0%) | 1/15 (6.67%) | 0/15 (0%) | 0/15 (0%) | 0/15 (0%) | 0/15 (0%) | 2/15 (13.33%) |
| 2017 | 12/52 (23.08%) | 18/52 (34.62%) | 1/52 (1.92%) | 2/52 (3.85%) | 5/52 (9.62%) | 5/52 (9.62%) | 9/52 (17.31%) | 33/52 (63.46%) |
| 2018 | 9/43 (20.93%) | 15/43 (34.88) | 22/43 (51.16%) | 1/43 (2.33%) | 0/43 (0%) | 1/43 (2.33%) | 4/43 (9.30%) | 30/43 (69.77%) |
| 2019 | 1/18 (5.56%) | 2/18 (11.11%) | 1/18 (5.56%) | 0/18 (0%) | 0/18 (0%) | 0/18 (0%) | 2/18 (11.11%) | 2/18 (11.11%) |
| 2020 | 47/307 (15.31%) | 63/307 (20.52%) | 73/307 (23.78%) | 16/307 (5.21%) | 35/307 (11.40%) | 11/307 (3.58%) | 52/307 (16.94%) | 174/307 (56.68%) |
| Total | 70/435 (16.09%) | 98/435 (22.53%) | 98/435 (22.53%) | 19/435 (4.37%) | 40/435 (9.19%) | 17/435 (3.91%) | 67/435 (15.40%) | 241/435 (55.40%) |

from $1.8 \times 10^1$ copies/$\mu$l to $1.8 \times 10^3$ copies/$\mu$l (Fig. S3). The reproducibility of the new PCR assays was verified using two dilutions ($1.8 \times 10^5$ copies/$\mu$l and $1.8 \times 10^3$ copies/$\mu$l) of the positive control by two individuals, which showed that all corresponding amplicons could be specifically produced (Fig. S4). All these results supported that a panel of PPV1–7 PCR assays with satisfied specificity, sensitivity and reproducibility were successfully developed.

**PPV prevalence in China.** The newly developed PPV1–7 PCR assays were further validated by detecting 435 tissue samples. The prevalence of each genotype of PPV in our field samples collected from 2016 to 2020 was shown in Table 1 and S2. Overall, 55.40% field samples (241 out of 435) were PPV positive, which were all confirmed by sequencing (Supporting data set). PPV2 and PPV3 were the most prevalent genotypes with both 22.53% (98/435), followed by PPV1 with 16.09% (70/435) and PPV7 with 15.40% (67/435), while the prevalence of PPV4-6 was less than 10% (PPV5 was 9.19% (40/435), PPV4 was 4.37% (19/435), PPV6 was 3.91% (17/435)). The overall prevalence of PPVs increased from 2016 (13.33%) to 2018 (69.77%), but declined at 2019 (11.11%) and then rebounded at 2020 (56.68%) in eight provinces of China. A decline of PPV prevalence in 2019 was likely associated with more strict management due to the outbreak of African swine fever in China since August 2018 (16). Remarkably, each genotype of PPVs could be detected from clinically healthy and diseased pigs (symptoms include fever, rash, meningitis, lymphadenopathy, hemorrhage, kidney pathology, and so on) (Table 2). In addition, the prevalence of PPV1–7 was higher in nursery pigs and finishing pigs than in suckling pigs (Fig. 1). More specifically, the prevalence of PPV1 (24.43%), PPV2 (30.53%), PPV3 (29.77%), PPV6 (6.87%), and PPV7 (20.61%) was highest in nursery pigs, while the prevalence of PPV4 (5.84%) and PPV5 (11.68%) was highest in finishing pigs. The lowest levels of PPV prevalence in nursery pigs suggested that there was likely no vertical transmission for PPVs.

**Coinfection status.** As shown in Table 3, simplex, duplex, triplex, quadruple, and quintuple infections were detected in 122, 83, 26, 7, and 3 field samples, respectively. No sample was detected to be infected by more than six genotypes of PPVs. There was 27.36% (119/435) samples were coinfected with at least two genotypes of PPVs. Noticeably, coinfection was detected in all eight provinces of China where our samples were collected. Geographic distribution analysis showed that all seven genotypes of PPVs could be detected in Shandong, Jiangsu and Fujian provinces, six genotypes could be found in Xinjiang and Anhui provinces, five genotypes could be identified in Guangdong province, and four genotypes could be determined in Hebei and Henan provinces (Fig. 2, Table S3). In addition, 166 out of the 435 samples were also PCV2 positive detected by our previously described method (17) (Table S2). The coinfection of PCV2 with PPVs was 22.30% (97/435). In details, the coinfections of PCV2 with PPV1 through PPV7 were 8.51% (37/435), 9.66% (42/435), 8.51% (37/435), 1.61% (7/435), 3.45% (15/435), 2.76% (12/435), and 6.67% (29/435), respectively. Even though the detection of different viral DNAs in a pig was not necessary to be active coinfection but also might be previous infection at different time points, the frequent co-detection of different viruses supported that the coinfections among different genotypes of PPVs or between PCV2 and PPVs occurred commonly in Chinese swine herds.

**PPV genome.** To evaluate the molecular characteristics and evolutionary trend of PPV1 through PPV7 in China in recent years, 20 representative PPV genomes (2 PPV1, 3

**TABLE 2** Twenty PPV genomes determined in this study

| No. | Name | Region[a] | Collection time | Genotype | GenBank no. | Symptoms |
|---|---|---|---|---|---|---|
| 1 | JSYZ20170418-30 | Yangzhou, Jiangsu | April 18, 2017 | PPV1 | MZ577026 | Death |
| 2 | SDQD20200424-830 | Qingdao, Shandong | April 24, 2020 | PPV1 | MZ577027 | Healthy |
| 3 | SDWF20171225-112 | Weifang, Shandong | December 25, 2017 | PPV2 | MZ577028 | Death |
| 4 | JSYZ20190725-717 | Yangzhou, Jiangsu | July 25, 2019 | PPV2 | MZ577029 | Rash, Pericardium pathology, Kidney pathology, Enteric pathology, Abdominal fluid, Meningitis |
| 5 | JSTZ20200503-1137 | Taizhou, Jiangsu | May 3, 2020 | PPV2 | MZ577030 | Healthy |
| 6 | HBTS20180519-151 | Tangshan, Hebei | May 19, 2018 | PPV3 | MZ577031 | Lymphadenopathy, Lung consolidation, Kidney pathology |
| 7 | HBTS20180519-155 | Tangshan, Hebei | May 19, 2018 | PPV3 | MZ577032 | Lymphadenopathy, Lung consolidation, Kidney pathology |
| 8 | SDQD20200424-829 | Qingdao, Shandong | April 24, 2020 | PPV3 | MZ577033 | Healthy |
| 9 | JSNJ20200425-838 | Nanjing, Jiangsu | April 25, 2020 | PPV3 | MZ577034 | Healthy |
| 10 | SDWF20170530-68 | Weifang, Shandong | May 30, 2017 | PPV4 | MZ577035 | Fever, Lung consolidation |
| 11 | FJFZ20200426-982 | Fuzhou, Fujian | April 26, 2020 | PPV4 | MZ577036 | Healthy |
| 12 | SDWF20170530-67 | Weifang, Shandong | May 30, 2017 | PPV5 | MZ577037 | Porcine dermatitis and nephropathy Syndrome (PDNS) |
| 13 | JSNJ20200426-911 | Nanjing, Jiangsu | April 26, 2020 | PPV5 | MZ577038 | Healthy |
| 14 | SDWF20170530-68 | Weifang, Shandong | May 30, 2017 | PPV6 | MZ577039 | Healthy |
| 15 | JSTZ20181121-431 | Taizhou, Jiangsu | November 21, 2018 | PPV6 | MZ577040 | High fever (42°C), kidney pathology, liver pathology |
| 16 | JSYZ20170103-20 | Yangzhou, Jiangsu | January 3, 2017 | PPV7 | MZ577041 | Death |
| 17 | SDWF20171204-106 | Weifang, Shandong | December 4, 2017 | PPV7 | MZ577042 | Healthy |
| 18 | HBTS20180519-152 | Tangshan, Hebei | May 19, 2018 | PPV7 | MZ577043 | Kidney pathology, lymph node hemorrhage |
| 19 | JSYZ20181025-377 | Yangzhou, Jiangsu | October 25, 2018 | PPV7 | MZ577044 | High fever (41°C), lymph node hemorrhage |
| 20 | JSYZ20190725-717 | Yangzhou, Jiangsu | July 25, 2019 | PPV7 | MZ577045 | Rash, Pericardium pathology, Kidney pathology, Enteric pathology, Abdominal fluid, Meningitis |

[a]The region indicated the city and province from where the sample was collected.

PPV2, 4 PPV3, 2 PPV4, 2 PPV5, 2 PPV6, and 5 PPV7) from both heathy and diseased pigs collected from different regions at different years were determined using primers shown in Table 4. The obtained nearly complete genomes were deposited into the GenBank database with accession numbers of MZ577026-MZ577045. The genome homologies between our obtained PPV1–7 genomes and corresponding representative PPV1–7 genomes were all higher than 90% (Fig. S5). The homologies for our PPV1 genomes and PPV1 Kresse strain (PPU44978) are ranged from 90.00 to 90.21%, our PPV2 and PPV2 BR/GO/ion strain (NC025965) are 95.60~97.24%, our PPV3 and PPV3 DJH24 strain (MK092412) are 95.06~97.08%, our PPV4 and representative PPV4 strain (NC014665) are 90.64~90.77%, our PPV5 and PPV5 IA469 strain (NC023020) are 91.73~91.89%, our PPV6 and PPV6 TJ isolate (NC023860) are 95.39~97.81%, our PPV7 and PPV7 GX49 isolate (NC040562) are 94.07~94.85%, respectively. However, the genome homologies among different genotypes of PPV genomes were less than 60.40% (PPV4 versus PPV5). Genome-based phylogenetic analysis showed that our PPV1 through PPV7 genomes were clustered with corresponding genotypes of PPVs (Fig. 3). The phylogenetic results were consistent with sequence comparisons and PCR results that they were PPV1 through PPV7, respectively.

**PPV recombination.** Our 20 PPV genomes and 194 PPV genomes available in GenBank were aligned and submitted to the recombination analysis. No recombination event was detected in our PPV1 through PPV6 genomes, however, all seven methods in RDP4 supported that PPV7 HBTZ20180519-152 strain is a natural recombinant virus with JX15 (MK092494) like isolate as the major parental virus and JX38 (MK092496) like isolate as the minor parental virus (Fig. 4A, Table S3). The JX38-like isolate provided the600–1648bp region within the ORF1 gene. Noticeably, our HBTZ20180519-152 strain was identified from the domestic pig but both JX15 and JX38 isolates were from wild boars. The crossover event could also be detected by SimPlot 3.5.1 (Fig. 4B).

**Polymorphic sites and selective pressure on VP2 of PPV1-7.** To evaluate the molecular characteristics of VP2 proteins, VP2 alignments were performed for each

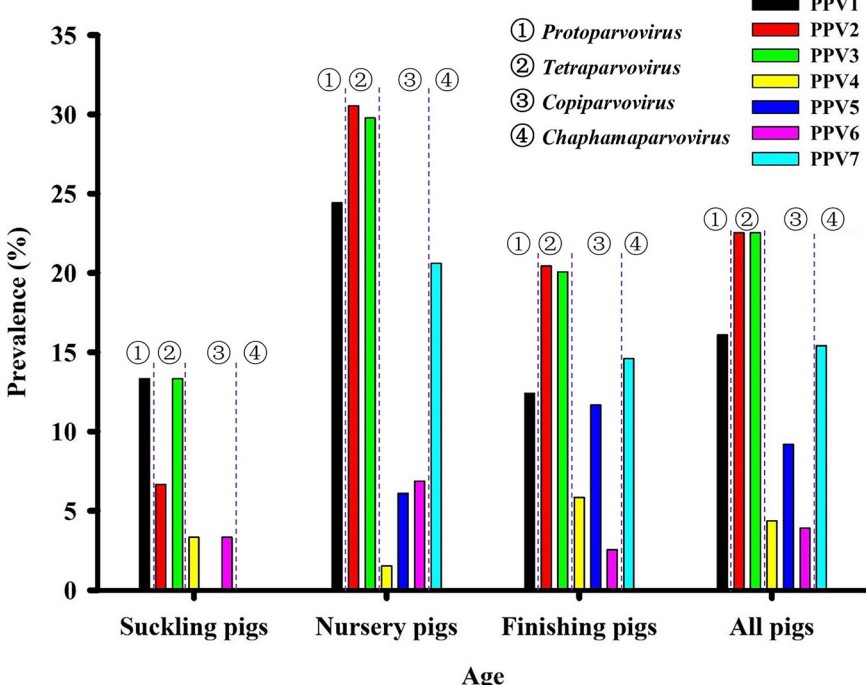

**FIG 1** The prevalence of PPV1–7 in different age groups. Overall, PPV2 and PPV3 were the most prevalent genotypes, followed by PPV1 and PPV7, while PPV4, PPV5 and PPV6 were less prevalent. In addition, the prevalence of PPVs was significantly higher in nursery pigs and finishing pigs than in suckling pigs. Specifically, the prevalence of PPV1, PPV2, PPV3, PPV6 and PPV7 were highest in nursery pigs, while the prevalence of PPV4 and PPV5 were highest in finishing pigs. The four genera, in which PPV1 through PPV7 were grouped, were shown.

genotype of PPVs using corresponding sequences from available complete genomes. Fig. 5 presented the PPV1 VP2 alignment result, which overall shared 99.62% amino acid identity (Fig. 5A) and contained several polymorphic sites (Fig. 5B). In details, the 320th residue of our PPV1 VP2 (JSYZ20170418-30 and SDQD20200424-830) was identical to the nonvirulent NADL2 strain but different from the highly virulent Kresse strain. The 45th, 407th, and 555th residues were identical to Kresse strain but different from NADL2 strain. The 228th, 414th, and 419th residues were identical to both NADL2 and Kresse strains but different from other Chinese isolates. The 144th, 215th, 436th, and 565th residues were different between our JSYZ20170418-30 and SDQD20200424-830 strains. The 144th, 215th, and 436th residues of JSYZ20170418-30 and the 565th residue of SDQD20200424-830 were identical to the Kresse strain. The VP2 alignments of each new genotype of PPV showed that the overall amino acid identities ranged from

**TABLE 3** Infection and coinfection of PPV1–7 in tissue samples

| Infection status | No. | Types |
|---|---|---|
| Simplex infection | 122 | PPV1 (31)[a], PPV2 (23), PPV3 (32), PPV4 (1), PPV5 (15), PPV6 (1), PPV7 (19) |
| Duplex infection | 83 | PPV1 + 2 (6), PPV1 + 3 (7), PPV1 + 4 (2), PPV1 + 7 (6), PPV2 + 3 (18), PPV2 + 4 (2), PPV2 + 5 (7), PPV2 + 6 (2), PPV2 + 7 (13), PPV3 + 4 (3), PPV3 + 5 (3), PPV3 + 7 (7), PPV4 + 5 (1), PPV4 + 6 (1), PPV4 + 7 (1), PPV5 + 6 (1), PPV5 + 7 (3) |
| Triplex infection | 26 | PPV1 + 2+3 (5), PPV1 + 2+4 (1), PPV1 + 2+5 (1), PPV1 + 2+7 (1), PPV1 + 3+4 (1), PPV1 + 3+6 (1), PPV1 + 3+7 (1), PPV1 + 4+7 (2), PPV2 + 3+4 (1), PPV2 + 3+5 (3), PPV2 + 3+6 (2), PPV2 + 3+7 (3), PPV2 + 5+6 (2), PPV3 + 5+7 (1), PPV3 + 6+7 (1) |
| Quadruplex infection | 7 | PPV1 + 2+3 + 7 (1), PPV1 + 2+5 + 7 (1), PPV2 + 3+4 + 7 (2), PPV2 + 3+6 + 7 (2), PPV3 + 4+5 + 6 (1) |
| Quintuple infection | 3 | PPV1 + 2+3 + 6+7 (2), PPV1 + 3+5 + 6+7 (1) |
| Sextuple infection | 0 | /[b] |
| Septuple infection | 0 | / |

[a]The numbers in parentheses are the sample numbers of each type of infection status.
[b]The diagonals indicated no detection.

**FIG 2** Coinfection of different genotypes of PPVs in eight provinces of China. All seven genotypes of PPVs could be detected in Jiangsu (JS), Fujian (FJ) and Shandong (SD) provinces. Six genotypes of PPVs could be detected in Xinjiang (XJ) and Anhui (AH). Five genotypes of PPVs could be found in Guangdong (GD). Four genotypes of PPVs could be found in Henan (HN) and Hebei (HB). The geographic distribution results indicated that different genotypes of PPVs are simultaneously prevalent in different regions of China.

94.37% to 99.70% and polymorphic sites were also determined in VP2 proteins of PPV2-7 (Fig. S6-S11).

Selective pressure analyses showed that the VP2 sequences of PPV1–7 were predominantly under negative pressure. The dN/dS ratios were ranged from 0.041 to 0.423 detected by FEL and from 0.0475 to 0.453 calculated by SLAC, respectively. A large amount of negative selection sites could be detected in VP2 sequences from PPV1 through PPV7, while only few positive selection sites (2, 7, and 1 detected by FUBAR, FEL, and SLAC) could be detected in VP2 sequences of PPV7 (Table 5).

**TABLE 4** Primers for PPV1–7 genome amplification

| Viruses | Primers | Sequences (5′ to 3′) | Location[a] | Length |
|---|---|---|---|---|
| PPV1 | PPV1-F1 | AGGTGGAGCCTAACACTATAAATACAGTTGCT | 4–79 | 1348bp |
| | PPV1-R1 | ATTGGCTGCATTGTAGCAACCA | 1374–1395 | |
| | PPV1-F2 | AGTCTGCCATGCTATAACTTGTGTACTAA | 1245–1273 | 1462bp |
| | PPV1-R2 | TCATTTCCTGTTGCAGACAATTCA | 2683-2706 | |
| | PPV1-F3 | TACATCTCAACAACCAGAGGTAAGAAGA | 2494–2521 | 1449bp |
| | PPV1-R3 | TTCCCTCCTATTGGATCTGAAGGTA | 3918–3942 | |
| | PPV1-F4 | ATGAACCAAATGGTGCTATAAGATTTACA | 3738–3766 | 986bp |
| | PPV1-R4 | CTAAAGACATAAGGTCATATAAGTGTGGTT | 4694–4723 | |
| PPV2 | PPV2-F1 | GCTTTCTAGTCGGACCGGAAGT | 1–22 | 1586bp |
| | PPV2-R1 | AGTGACCTTGAGCATGCGGC | 1567-1586 | |
| | PPV2-F2 | AAGGGAAGATGTCTGAGAAGTTTGTAGA | 1375–1402 | 1729bp |
| | PPV2-R2 | CTATATCCAGGCAGGGTGAGTCCT | 3081–3104 | |
| | PPV2-F3 | CAGCCGGCACCTGAAGAGA | 2918–2936 | 1424bp |
| | PPV2-R3 | CGGAGGTTCAACGTGCCC | 4324–4341 | |
| | PPV2-F4 | CTGAGTATTCAGTGGTACAACCCTCC | 4097–4122 | 1405bp |
| | PPV2-R4 | AGGTGAGTCAGCACTTCCGGA | 5481–5501 | |
| PPV3 | PPV3-F1 | CGGTTCCGGTTGTGACGTC | 1–19 | 1574bp |
| | PPV3-R1 | TGGCAACGGAGCTAGGCC | 1557–1574 | |
| | PPV3-F2 | TCAGAGGATTTTATACCTACCTGTGTCA | 1410–1437 | 1593bp |
| | PPV3-R2 | TCGTGGTGTTTCGCTGCTTC | 2983–3002 | |
| | PPV3-F3 | AGCCGATAGCTACGGTGGCA | 2823–2842 | 1291bp |
| | PPV3-R3 | TCCAGAATTATCTACCCCTGTCATGA | 4088–4113 | |
| | PPV3-F4 | TGGGAATTGCTGACCATAGGTC | 3902–3923 | 1283bp |
| | PPV3-R4 | ACACAATTCCGGTTCCGGTT | 5165–5184 | |
| PPV4 | PPV4-F1 | TGTAAATTTATYTATGCAAAGTAGGAGGA | 1–29 | 1395bp |
| | PPV4-R1 | CGTTTGTTTCATGCTCCAAAAGTA | 1372–1395 | |
| | PPV4-F2 | AGAATCGCTCAGCAAAAGTTATTGA | 1189-1213 | 1587bp |
| | PPV4-R2 | TAAACACATATTCTGGTTCTTGCTGAAT | 2748—2775 | |
| | PPV4-F3 | ATGGAGAACAGAAAGCAAACTGAGAT | 2552–2577 | 1618bp |
| | PPV4-R3 | TTGGATGTCCTGGTCCTTGATCT | 4147–4169 | |
| | PPV4-F4 | TGGTGGTCAAGTATCTGTGCCA | 4021–4042 | 1392bp |
| | PPV4-R4 | CAATAAAGAGGAAGTCTTTTTTTTAACTTCA | 5384–5412 | |
| PPV5 | PPV5-F1 | TCAAAGGTCACTTCCGGGTCA | 62–82 | 1289bp |
| | PPV5-R1 | AGAACAAATATTATTGTCCACCAACCA | 1324–1350 | |
| | PPV5-F2 | ATTGGAGGTACAATAGGTGAAGCCT | 1177–1201 | 1612bp |
| | PPV5-R2 | ACTTTCATATATGGAAACGCAGCTTC | 2763–2788 | |
| | PPV5-F3 | TTTGCTTGCCAGAATAGCTGATTT | 2606–2629 | 1545bp |
| | PPV5-R3 | ACTGTGTCACAGCATTGTTTCCCT | 4127–4150 | |
| | PPV5-F4 | CATATACCTAGCCTTGTTGTCTGTACTCC | 3951–3979 | 1557bp |
| | PPV5-R4 | TTATCTTCTCGCTCTAACACGTTGCT | 5482–5507 | |
| PPV6 | PPV6-F1 | GTGCACAAAATAAAAGTGTGATGTTTTC | 1–28 | 1629bp |
| | PPV6-R1 | ACATTCAACCACATCCTCTGTCATCT | 1604–1629 | |
| | PPV6-F2 | CCAAACGGGTAAGAGAAACAGTATTT | 1428–1453 | 1657bp |
| | PPV6-R2 | ATATAGGCTAGCATATAGGGGATCTGCT | 3057–3084 | |
| | PPV6-F3 | AGATCGGTGAGAGTGTTTCATGGA | 2853–2876 | 1587bp |
| | PPV6-R3 | AACAGTATTTATCAGGCCAGGGTGA | 4415–4439 | |
| | PPV6-F4 | TCCAGTAGACCCATCTGCCACC | 4180–4201 | 1645bp |
| | PPV6-R4 | ATATTGATTGATGAAGGTCTGACCAGA | 5798–5824 | |
| | PPV6-F5 | CCTGCCTGTTTAGAGTCCCAGAT | 5624–5646 | 765bp |
| | PPV6-R5 | GCTTCGCGGCCCATGC | 6373—6388 | |
| PPV7 | PPV7-F1 | GGAACGACAAGGACGACACTTC | 1–22 | 1619bp |
| | PPV7-R1 | CGCGCACTCCGGACTGAT | 1602—1619 | |
| | PPV7-F2 | ACAACTTCGACGCGACCATGTA | 1393—1414 | 1557bp |
| | PPV7-R2 | AGACCTTCTTTGTAGGCCACGTACC | 2925–2949 | |
| | PPV7-F3 | CACCACCGTCTTCAATCCCATA | 2764–2785 | 1236bp |
| | PPV7-R3 | TGGCGTTGAGAAGACACTGGTTTA | 3976–3999 | |

[a]The location of each primer was determined based on the comparison of 11 PPV1 genomes, 28 PPV2 genomes, 7 PPV3 genomes, 15 PPV4 genomes, 46 PPV5 genomes, 38 PPV6 genomes, and 49 PPV7 genomes available from the GenBank database, respectively.

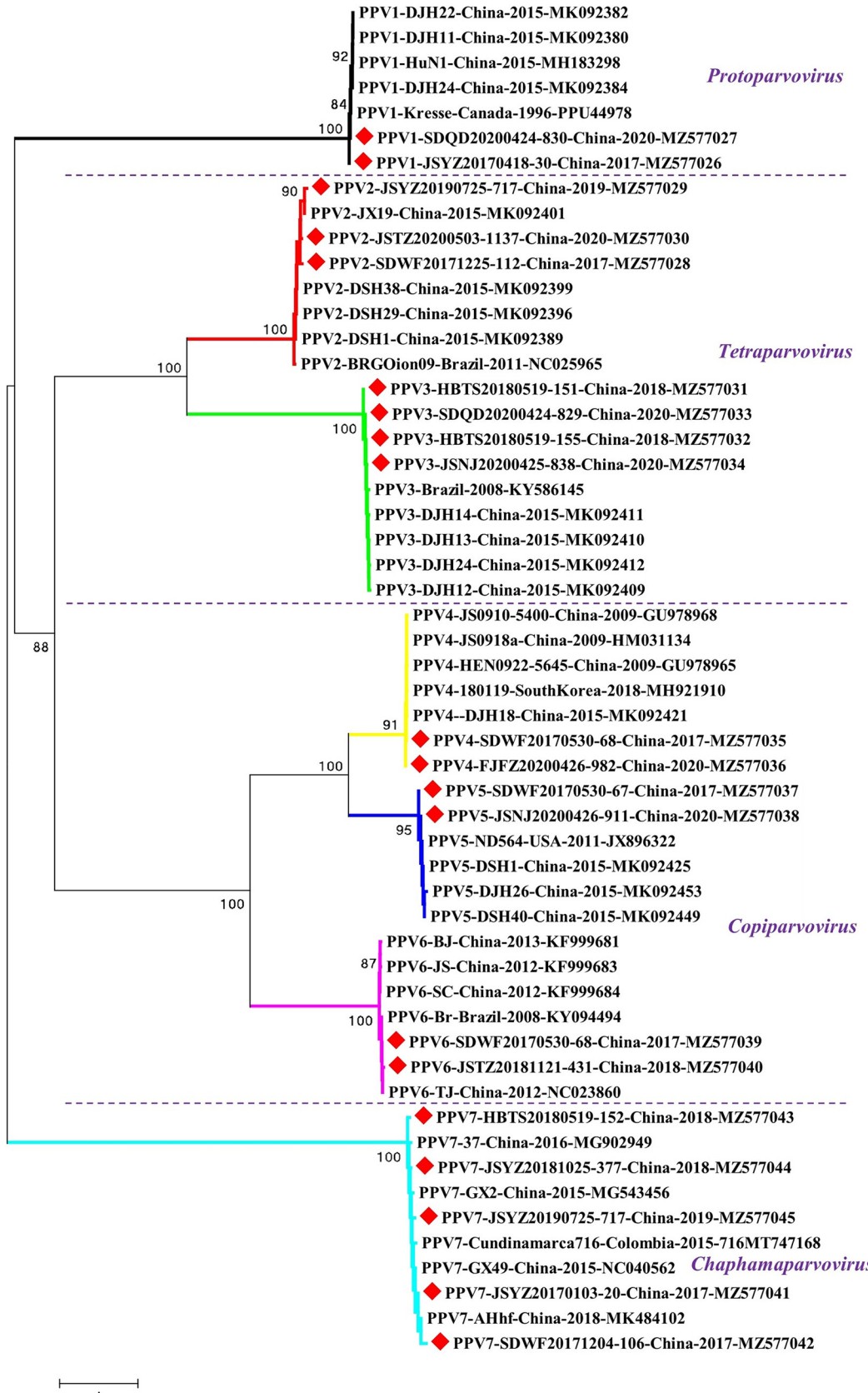

**FIG 3** Genome-based genotyping based on 20 PPV genomes obtained in this study and 35 representative PPV genomes (five for each genotype) from GenBank database using MEGA 6.06. Different genotypes were shown in different colors.

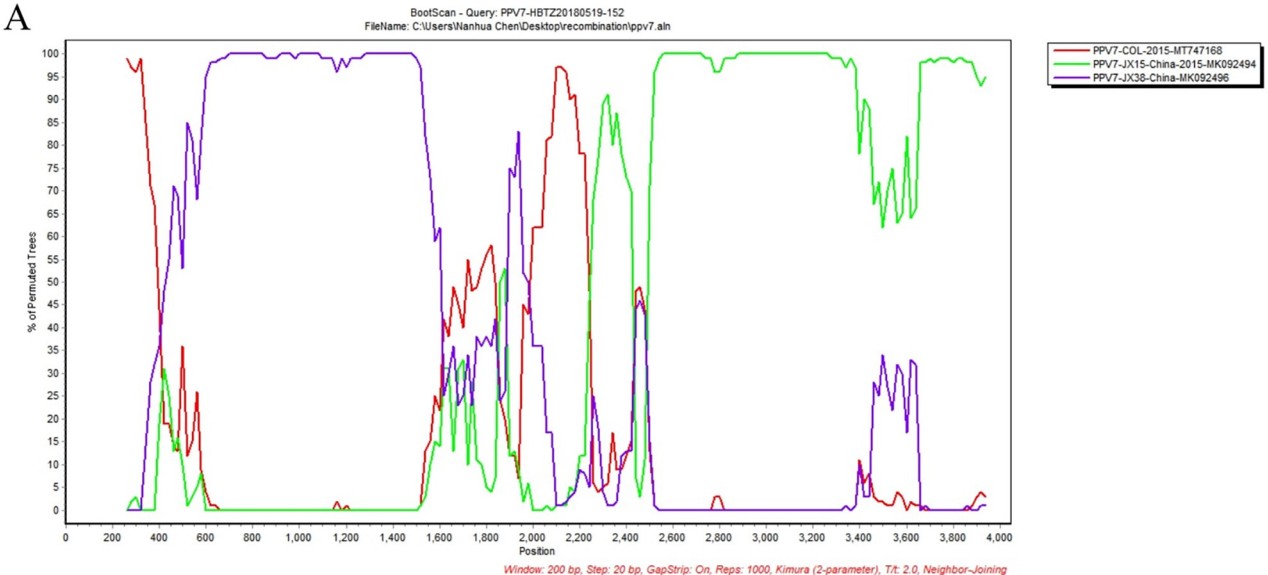

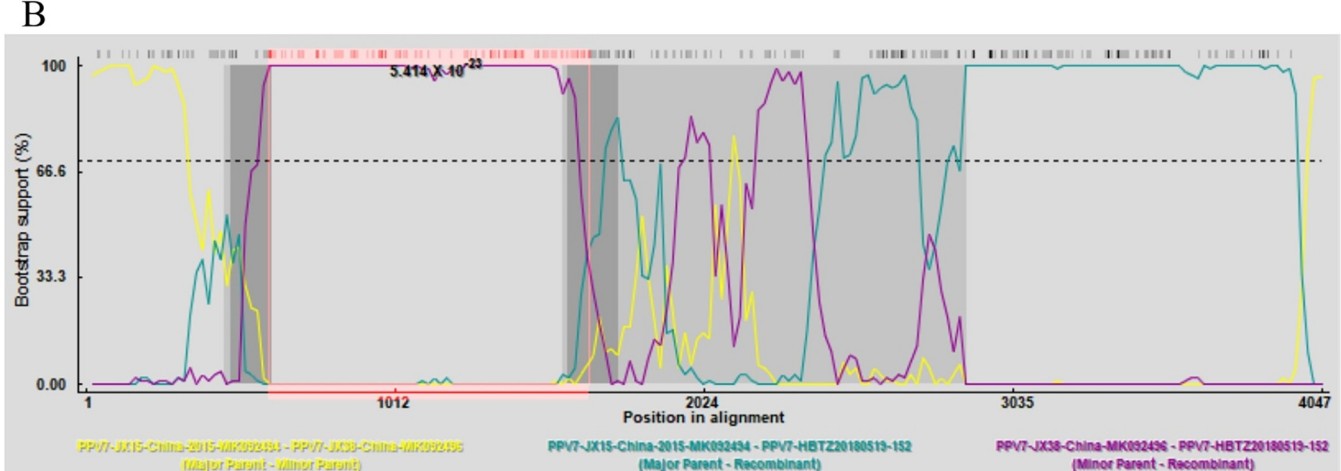

**FIG 4** The crossover event in PPV7 HBTZ20180519-152 genome was detected by both SimPlot and RDP4. In SimPlot analysis (A), the query HBTZ20180519-152 strain was recombined from parental JX15-like and JX38-like isolates. The y axis shows the percentage of permutated trees employing a sliding window of 200 nucleotides (nt) and a step size of 20 nt. The other options, including Kimura (2-parameter) distance model, 2.0 Ts/Tv ratio, Neighbor-Joining tree model, 1000 Bootstrap replicates were used. In RDP4 analysis (B), the crossover event could also be detected by all seven methods embedded in RDP4 (Table S4).

## DISCUSSION

Since the emergence of new PPVs in China, majorities of the studies focus on the genetic characterization of individual isolates and prevalence of one or few genotypes of PPVs in one area (7, 10, 12–14, 18). The up-to-date epidemiology of the seven genotypes of PPVs in China is of great importance but is unclear in recent years. This systematic investigation unveiled the prevalence, coinfection, and evolution of PPV1–7 in China. In addition, the molecular characteristics of PPV1–7 prevalent in Chinese swine herds in the last 5 years were also determined.

Singular traditional PCR and real-time PCR assays have been developed for all seven genotypes of PPVs (19–22). Recently, a multiple PCR assay for PPV1–7 has also been

**FIG 3** Legend (Continued)

The four genera in which each PPV genotype was clustered were also shown. Consistent with sequence alignments and PCR results, our PPV strains were clustered with corresponding genotype of PPVs in the phylogenetic tree. Our PPV strains are highlighted with red diamonds. Each virus is presented by the genotype, virus name, isolation country, isolation year and the GenBank accession number. Bootstrap values from 1000 replications are indicated for each node.

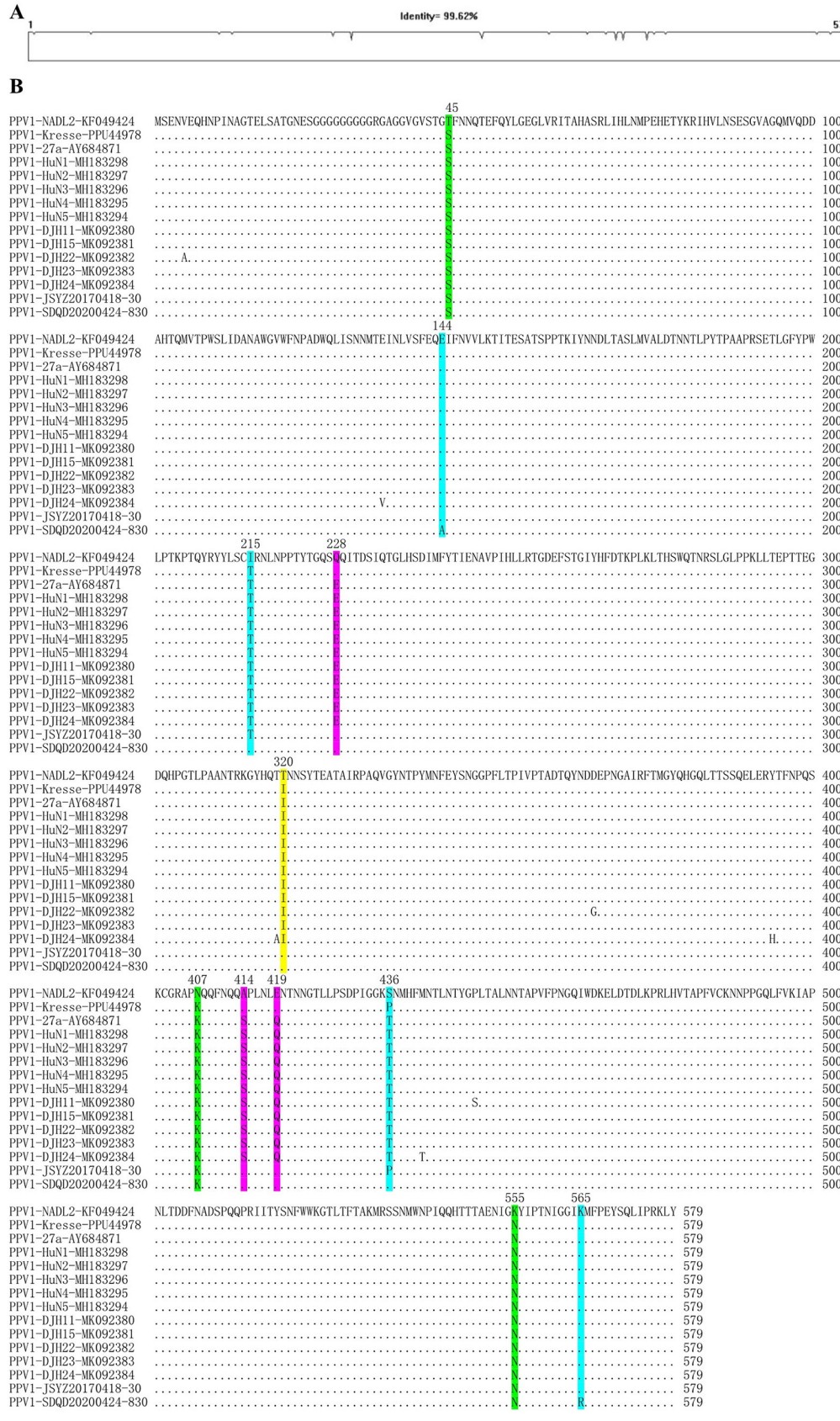

**FIG 5** Sequence comparison of PPV1 VP2 proteins. Overall similarity of PPV1 VP2 sequences (A). Polymorphic sites in the VP2 are marked in distinct colors (B).

**TABLE 5** Selection pressure analyses for the VP2 proteins of PPVs

| Test method | Virus | No. of codons | dN/dS | Positive selection | Negative selection | Criteria |
|---|---|---|---|---|---|---|
| FUBAR | PPV1 | 579 | | 1 | 12 | Posterior probabilities of 0.9 |
| | PPV2 | 536 | | 0 | 185 | |
| | PPV3 | 555 | | 0 | 58 | |
| | PPV4 | 541 | | 0 | 86 | |
| | PPV5 | 602 | | 0 | 29 | |
| | PPV6 | 594 | | 3 | 71 | |
| | PPV7 | 474 | | 2 | 191 | |
| FEL | PPV1 | 579 | 0.423 | 0 | 2 | P value threshold of 0.05 |
| | PPV2 | 536 | 0.0872 | 0 | 75 | |
| | PPV3 | 555 | 0.0914 | 0 | 18 | |
| | PPV4 | 541 | 0.041 | 0 | 30 | |
| | PPV5 | 602 | 0.224 | 0 | 20 | |
| | PPV6 | 594 | 0.0800 | 0 | 28 | |
| | PPV7 | 474 | 0.116 | 7 | 151 | |
| SLAC | PPV1 | 579 | 0.453 | 0 | 0 | P value threshold of 0.05 |
| | PPV2 | 536 | 0.104 | 0 | 28 | |
| | PPV3 | 555 | 0.107 | 0 | 1 | |
| | PPV4 | 541 | 0.0475 | 0 | 3 | |
| | PPV5 | 602 | 0.233 | 0 | 3 | |
| | PPV6 | 594 | 0.103 | 0 | 6 | |
| | PPV7 | 474 | 0.244 | 1 | 89 | |

reported (5). In this study, we developed a panel of PPV1–7 detection assays. The sensitivities of our assays are same or 10-fold higher than previous PCR assays (5, 19). Even though the PPV1–7 amplicons were also designed with distinguishable sizes, however, due to the interference among different primer sets, we failed to develop the multiple PCR assay for PPV1–7 simultaneous detection using the panel of primers in this study, indicating that further optimizations are needed. Even so, we provided alternative PCR assays for PPV1 through PPV7 detection in this study.

The prevalence of each genotype of PPV in China has been widely examined but the prevalent percentages are distinct. It is around 5.56–30% for PPV1 (23, 24), 2.4–39.56% for PPV2 (4, 24), 45.11–46.5% for PPV3 (4), 0.76 to 21.56% for PPV4 (4, 13, 24), 3.8–75% for PPV6 (10), 32.8% for PPV7 (15). PPV5 was also detected in China but its prevalence was not determined (14). In United States, the prevalence of PPV5 was 2.6–6.6%, which was slightly higher than PPV4 (1.9–4.1%) (8). Consistent with previous studies, our results also supported that PPV1 through PPV7 are widely prevalent in different areas of China. Furthermore, our results presented the up-to-date infection status of each genotype of PPVs in Chinese swine herds.

The coinfection of PPV with PCV2 was frequently reported. A previous study showed that coinfection of PCV2 and PPV was 14.3% in the serum samples and 49.4% in the tissue samples (25). A recent study showed that the coinfection of PPV2 and PCV2 was 12.9%, but the PCV2 and PPV3 coinfection and the PCV2 and PPV4 coinfection were only 3.6% and 2.88%, respectively (26). The coinfections of PCV2 and PPVs in our tissue samples were all less than 10%. The highest coinfection was between PPV2 and PCV2, which is consistent with the recent report (26). In addition, the coinfections among different genotypes of PPVs were also frequently detected in this study. A previous study showed that both PPV1 and PPV7 may stimulate the replication of PCV2 (27). Considering that our study and previous reports confirmed the detection of each genotype of PPVs in both clinically healthy and seriously diseased pigs (6, 7, 10–12, 14, 26), the coinfections among PPVs or with PCV2 might have synergistic effects on the PPV1 associated SMEDI syndrome or the PCV2 associated PCVAD.

Previous studies have determined one or few (nearly) complete genomes of few genotypes of PPVs in a pig farm or a targeting region (14, 18, 28). To systematic

investigate the up-to-date evolution of PPVs in Chinese swineherds, 20 PPV nearly complete genomes have been determined in this study. Genome based phylogenetic analysis and genome comparison suggested that the genetic diversity of Chinese PPVs did not significantly increase in the last 5 years.

Recombination is one of the evolutionary mechanisms that are crucial for the generation of genomic diversity. Recombination seems to play an important role in PPV evolution because several chimeric strains have been detected (29). PPV1 2074-7 isolate is a recombinant from the parental Kresse strain and IDT isolate, whereas the PPV1 225b isolate is a recombinant from the parental 27a and IDT strains (30). PPV2 recombination events in VP genes within and between country strain recombinants were first reported in 2013 (4). PPV7 KF4 strain was identified as a recombinant with 17KWB09 strain from Korean wild boar as the major parental virus and N133 strain from Korean domestic pigs as the minor parental strain (31). Both RDP4 and Simplot analyses supported that our PPV7 HBTZ20180519-152 strain from Chinese domestic pig in 2018 is a recombinant from the major parental JX15-like virus and the minor parental JX38-like strain both isolated from Chinese wild boars in 2015. The identification of PPV7 intraspecies recombinant in this study suggested that PPV7 natural recombination occurred in wild boars and might be beneficial for the transmission of PPV from wild boars to domestic pigs.

The VP2 of PPV1 was considered to contain virulence determinants and the influence of polymorphic sites have been evaluated (32–34). Five substitutions (I215T, D378G, H383Q, S436P, R565K) observed in virulent Kresse strain are likely associated with the pathogenicity (32). The 378, 383 and 565 residues might be involved in the immune response, while the 436 residue located in the 3-fold spike center of the capsid subunit seems to be a critical antigenicity associating site (32, 33, 35). The amino acid substitutions in the 3-fold spike region are likely related to the affinity of neutralizing antibodies (33, 34). Antisera from pregnant sows infected with various PPV1 strains showed high neutralizing activities against homologous viruses but much lower neutralizing activity against heterologous PPV1 27a isolate (34). Considering the low neutralizing activity against same genotype but heterologous isolate, cross-neutralizations among different genotypes of PPVs even though were not determined but are likely very limited. Our PPV1 isolates not only contain substitutions identical to nonvirulent NADL2 strain and/or virulent Kresse isolate, but also comprise some new substitutions. In light of complex VP2 substitution patterns of our PPV isolates, the exact influence of these residues on pathogenicity, cross-neutralization or neutralization escape deserves further investigation.

The dN/dS ratios in VP2 sequences of PPV1–7 were all lower than 0.5 by different detection methods, indicating that most sites in VP2 of PPV1–7 are under strong negative selection. However, positive selection sites were also detected in VP2 of PPV7. Our results are similar to the previous studies (2, 36). Considering that the VP2 of PPV7 has the highest diversity comparing with the corresponding region of other genotypes of PPVs, these results supported that selective pressure and evolutionary rate are correlated (37). The pressure selected variable sites and nonsynonymous mutations are likely related to antigenic alternations in response to the host immune responses (2, 36).

In conclusion, our findings provide the first systematic investigation on the prevalence, coinfection and evolution of PPV1 through PPV7 in Chinese swine herds from 2016 to 2020, which are beneficial for a better understanding of PPV1 through PPV7 epidemiology in Chinese swine herds.

## MATERIALS AND METHODS

**Virus strains and clinical samples.** All swine viruses used in this study were stored in our laboratories, including porcine circoviruses (PCV) JSNJ2004-913 (PCV1), JSYZ1705-32 (PCV2), SD17-36 (PCV3), and JSYZ1901-2 (PCV4) strains, pseudorabies virus (PRV) XJ03 strain, porcine epidemic diarrhea virus (PEDV) XM2-4 strain, classical swine fever virus (CSFV) JS1805-2 strain, porcine reproductive and respiratory syndrome virus (PRRSV) HLJB1 strain (PRRSV1) and SD17-38 isolate (PRRSV2) (38–41). These swine viruses were used in the specificity tests of the panel of PPV1–7 PCR assays. A total of 435 pig tissue samples (including lungs, lymph nodes or tonsils depended on clinical samples available) were submitted from eight provinces (Shandong, Hebei, Henan, Jiangsu, Anhui, Fujian, Guangdong and Xinjiang) to the Animal Hospital at Yangzhou University from April 2016 to May 2020.

**Primer design and viral nucleic acid extraction.** Multiple sequence alignments were carried out using all 194 PPV complete genomes (containing 11 PPV1, 28 PPV2, 7 PPV3, 15 PPV4, 46 PPV5, 38 PPV6, and 49 PPV7) available from the GenBank database. Primers were designed within the highly conserved regions specific for each genotype of PPVs (Table S1). The specificity of all primers was validated by multiple comparisons with the DNAMAN software and the BLAST tool (https://blast.ncbi.nlm.nih.gov/Blast.cgi).

Viral nucleic acids were extracted as we previously described (42). Briefly, viral DNAs were extracted from tissue samples using HiPure Tissue DNA minikit (Magen, Guangzhou, China). Viral RNAs used for specificity tests of the newly developed assays were extracted from the supernatants of virus-infected cell cultures using the TRIpure Reagent according to the manufacturer's instructions (Aidlab, Beijing, China). First-strand cDNAs were synthesized using the extracted total RNAs and PrimeScript 1st Strand cDNA Synthesis kit (TaKaRa, Dalian, China). Viral nucleic acids were eluted with 30 $\mu$l of nuclease-free double distilled water (ddH$_2$O) and stored at $-80°C$ until used.

**Construction of standard positive control.** To prepare the standard positive control, seven amplicons from the panel of PCR assays were combined as a standard gene for synthesis (Genewiz, Suzhou, China) as we described previously (43). The synthetic sequence was cloned into the pUC57 vector and transformed into DH5$\alpha$ chemically competent cells. The recombined plasmid was purified with the HiSpeed Plasmid minikit (Qiagen, Germany) and quantified by measuring OD260 with the spectrophotometer Nano-200 (Aosheng, Hangzhou, China). The concentration was converted into copy numbers using the following formula: y (copies/$\mu$l) = $(6.02 \times 10^{23}) \times$ (x [ng/$\mu$l] $\times 10^{-9}$ DNA)/(DNA length $\times 660$) (44). The plasmid was diluted with ddH$_2$O to obtain the stock solution containing $10^8$ copies of the plasmid DNA per microliter.

**Establishment of PPV1–7 PCR assays.** Primer concentrations and amplification conditions were optimized according to our previous studies (44, 45). After optimization, each of the PCR assays was carried out in the 20 $\mu$l reaction system containing 2 $\mu$l DNA, 0.5 $\mu$l primers (10 $\mu$M) and 10 $\mu$l Premix *Taq* (TaKaRa, Dalian, China). The amplification was performed at 30 cycles of 98°C for 10s, 55°C for 30s, and 72°C for 1 min according to the manufacturer's instructions. The PCR products were analyzed on 1.5% agarose gel electrophoresis in 1$\times$ TAE buffer.

**Validation of PPV1–7 detection assays.** To evaluate the specificity of the new PCR assays, DNA from seven genotypes of PPV (PPV1 to PPV7) positive samples and DNA/cDNA from the other viruses (PCV, PRV, PRRSV, CSFV, PEDV) were tested. The sensitivity of the assays was determined using 10-fold serially diluted standard plasmids (101–107 copies/$\mu$l). The reproducibility of the assays was tested by using two dilutions of the positive-control plasmid by two individuals. To further validate the panel of PCR assays, 435 clinical samples were detected by all the new assays. The amplicon from each positive sample was confirmed by sequencing (Tsingke Biotechnology Co., Ltd.).

**PPV genome determination and phylogenetic analysis.** Representative PPV positive samples were submitted to complete genome sequencing using primers amplifying overlapped fragments as shown in Table 4. The sequences were assembled by the DNAMAN 6.0 software to generate 20 nearly complete genomes (Table 2). To evaluate the evolutionary relationships between our PPV genomes and other PPV genomes, the obtained 20 complete genomes and 35 representative PPV genomes (5 genomes for each genotype) were aligned by the ClustalX 2.0. Genome-based phylogenetic trees were constructed using MEGA 6.06 as we previously described (40). In details, phylogenetic analysis was performed based on the aligned sequence using the neighbor-joining method and the maximum composite likelihood model. The robustness of the phylogenetic tree was evaluated by bootstrapping using 1000 replicates.

**Recombination detection.** To evaluate the role of recombination in the generation of our PPV strains, the multiple genome alignment based on our 20 genomes and all 194 PPV genomes available from GenBank was submitted to screen potential recombination events by recombination detection program 4 (RDP4) (46, 47). Briefly, seven methods embedded in RDP4 software, including RDP, GENECONV, BootScan, Maxchi, Chimaera, SiScan, and 3Seq, were used to detected the recombination events and breakpoints. The default settings were used for all the seven methods, and the highest acceptable *P value* was set at 0.05. In addition, the detected recombination events were further confirmed by SimPlot 3.5.1 as described previously (39, 48).

**Selective pressure analyses.** The VP2 coding sequences from PPV1 through PPV7 genomes obtained in this study and available genomes from the GenBank database were used for selective pressure analyses. The selective pressure was evaluated through the ratio dN/dS (the ratio of the nonsynonymous evolutionary rate (dN) to the synonymous evolutionary rate (dS)) (49). It was considered diversifying (positive), neutral and purifying (negative) selection when the dN/dS ratio was higher, equal and less than 1, respectively. The alignment was evaluated for selection using A Fast, Unconstrained Bayesian AppRoximation for Inferring Selection (FUBAR), Fixed Effects Likelihood (FEL) and Single-Likelihood Ancestor Counting (SLAC) methods provided by the Data-Monkey Web Server (http://www.datamonkey.org) (50–52). Sites were considered under selective pressure when detected by at least two methods. The significance level was set to a posterior probability higher than 0.9 for FUBAR and $P < 0.05$ for FEL and SLAC as previously described (53).

## SUPPLEMENTAL MATERIAL

Supplemental material is available online only.

**SUPPLEMENTAL FILE 1**, PDF file, 4 MB.

**SUPPLEMENTAL FILE 2**, XLSX file, 0.04 MB.

## ACKNOWLEDGMENTS

This work is funded by National Natural Science Foundation of China (31802172) and the Priority Academic Program Development of Jiangsu Higher Education Institutions (PAPD). Nanhua Chen is supported by the Natural Science Foundation for Excellent Young Scholars of Jiangsu Province (BK202111603) and High Talent Supporting Program of Yangzhou University.

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
