## [Reviewer comments · Microbiology Spectrum]

Microbiology Spectrum

A systematic investigation unveils high co-infection status of porcine parvovirus types 1 through 7 in China from 2016 to 2020

Jixiang Li, Yanzhao Xiao, Ming Qiu, Xinshuai Li, Shubin Li, Hong Lin, Xiangdong Li, Jianzhong Zhu, and Nanhua Chen

Corresponding Author(s): Nanhua Chen, Yangzhou University

Review Timeline:

Submission Date:	August 16, 2021
Editorial Decision:	September 26, 2021
Revision Received:	October 12, 2021
Accepted:	October 29, 2021

Editor: Clinton Jones

Reviewer(s): The reviewers have opted to remain anonymous.

Transaction Report:

DOI: <https://doi.org/10.1128/Spectrum.01294-21>

September 26, 2021

Dr. Nanhua Chen
Yangzhou University
Preventive Veterinary Medicine
48 East Wenhui Road
R404 International Cooperation Building
Yangzhou, Jiangsu 225009
China

Re: Spectrum01294-21 (A systematic investigation unveils high co-infection status of porcine parvovirus types 1 through 7 in China from 2016 to 2020)

Dear Dr. Nanhua Chen:

Thank you for submitting your manuscript to Microbiology Spectrum. When submitting the revised version of your paper, please provide (1) point-by-point responses to the issues raised by the reviewers as file type "Response to Reviewers," not in your cover letter, and (2) a PDF file that indicates the changes from the original submission (by highlighting or underlining the changes) as file type "Marked Up Manuscript - For Review Only". Please use this link to submit your revised manuscript - we strongly recommend that you submit your paper within the next 60 days or reach out to me. Detailed information on submitting your revised paper are below. Please pay particular attention to Reviewer #2 who pointed out that there was significant redundancy in certain sections in the Results and Discussion. Please pay particular attention to remove the redundant features of these sections.

Link Not Available

Sincerely,

Clinton Jones

Journals Department
Reviewer comments:

Reviewer #1 (Comments for the Author):

This is an interesting retrospective study about porcine parvovirus infections in china from 2016-2020. Besides reporting infections with the "old" genotype PPV-1, this manuscript not only documents infections with six more recently described genotypes, PPV2 through PPV-7, but also convincingly documents co-infections with different genotypes, and strongly suggest evolutionary drift through recombination. Indeed, the manuscript presents a comprehensive data-set, including 20 almost complete sequences of representative isolates of the different genotypes.

Although the manuscript is clearly structured, the magnitude of the data set covering two subfamilies with four genera infecting pigs, makes it in some aspects rather difficult to follow. Thus, it could be beneficial to include some modifications in order to improve the visibility of some issues.

Specific comments:

Table S1 should include the position of the primers within the genome and should be added to the first paragraph of the results section rather than Supplementary information.

Figure 1 and 2nd Paragraph of the result section. It would be helpful to associate the different genotypes to their respective Parvovirus genera/subfamilies providing more visibility about the relationship of the different genotypes. Are the differences according to the prevalence of PPV1-7 to the different Age-Groups statistically relevant and is the absence of PPV7 (and potentially PPV5) in suckling pigs conclusive?

Figure 2: It would be of interest to additionally indicate the genotypes (genera/subfamilies) detected in the corresponding areas of China.

In regard to the PPV genome analyses it would be interesting to know whether the variable regions are clustering toward a distinct place within the genome.

Although of minor importance it would make sense to color-code the PPV genotypes in Fig. 3 the same as in Fig. 1.

Figure 5 it would be worth indicating changes towards NADL2 (non-virulent) to Kresse (virulent) strains as discussed Lanes 199ff. Are the changes towards the one or the other in agreement with the described pathology? Maybe it would be worthwhile to include such information into this figure.

Considering the diversity of the different genotypes regarding their genera/subfamily of Parvoviridae it is of interest to comment on the serotypes and potentially cross-neutralizations vs. neutralization escape. If there are valid information in this regard it would be of interest to include this in the discussion.

The numbering of the tables does not really follow the flow of the manuscript. Thus Table 1 is part of Mat/Met, Table 3 is preceding Table 2 in paragraph 2 of the results section. This make it difficult to follow.

Reviewer #2 (Comments for the Author):

A study of the porcine parvoviral DNA in swine in different regions of China, using PCR to detect the PPV 1-7 sequences. For each virus type some were sequenced, and the sequences analyzed by standard methods (phylogenetics, recombination analysis, and sequence selection). This is very standard in the field and the data is similar to those that have been reported previously in China and other regions of the world, although with more details and some additional sequences. There are no specific surprises here - it has long been known that multiple virus strains co-circulate in swine (and most other animals), and that there is recombination between parvoviruses.

There are some issues that should be addressed:

- 1) It should be made clear that the detection of DNA in the samples is not necessarily related to a current active co-infection, but the DNA is likely to be left over from a previous infection of that animal.
- 2) The recombination is not of the exact viruses listed, but of viruses with sequences related to those viruses.
- 3) There is a lot of repetition between the Results and Discussion - I suggest that the Discussion be rewritten to highlight the main new discoveries, but not of over all of the details again. Likewise, the summary and conclusions largely repeat the Abstract, and can be mostly removed.

Staff Comments:

Preparing Revision Guidelines

Please return the manuscript within 60 days; if you cannot complete the modification within this time period, please contact me. If

you do not wish to modify the manuscript and prefer to submit it to another journal, please notify me of your decision immediately so that the manuscript may be formally withdrawn from consideration by Microbiology Spectrum.

Dear editor,

Thank you for your kind consideration of our manuscript. Our manuscript has been revised according to the reviewers' comments point-by-point. The details are shown as follows:

Reviewer #1 (Comments for the Author):

This is an interesting retrospective study about porcine parvovirus infections in china from 2016-2020. Besides reporting infections with the "old" genotype PPV-1, this manuscript not only documents infections with six more recently described genotypes, PPV2 through PPV-7, but also convincingly documents co-infections with different genotypes, and strongly suggest evolutionary drift through recombination. Indeed, the manuscript presents a comprehensive data-set, including 20 almost complete sequences of representative isolates of the different genotypes.

Although the manuscript is clearly structured, the magnitude of the data set covering two subfamilies with four genera infecting pigs, makes it in some aspects rather difficult to follow. Thus, it could be beneficial to include some modifications in order to improve the visibility of some issues.

Reply: Thank you for your kind consideration of our manuscript. The manuscript has been revised according to your suggestions. The detailed revisions are as follows.

Specific comments:

Table S1 should include the position of the primers within the genome and should be added to the first paragraph of the results section rather than Supplementary information.

Reply: Thank you for your suggestion. The table S1 has been revised and included in the first paragraph of the results (line 116).

Figure 1 and 2nd Paragraph of the result section. It would be helpful to associate the different genotypes to their respective Parvovirus genera/subfamilies providing more visibility about the relationship of the different genotypes. Are the differences according to the prevalence of PPV1-7 to the different Age-Groups statistically relevant and is the absence of PPV7 (and potentially PPV5) in suckling pigs conclusive?

Reply: Thank you for your suggestion. To improve the visibility about the relationship of the seven PPVs, corresponding genus for each PPV has also been shown in Figure 1. As we could notice in figure 1, the PPV2 and PPV3 belonging to *Tetraparvovirus* genus are most prevalent, while the PPV4, PPV5 and PPV6 clustering in *Copiparvovirus* genus are the least detected in our samples. However, due to small amount of samples (only 30) from suckling pigs, we don't think it's

conclusive that PPV5 and PPV7 are absent in suckling pigs.

Figure 2: It would be of interest to additionally indicate the genotypes (genera/subfamilies) detected in the corresponding areas of China.

Reply: Thank you for your suggestions. The different genotypes of PPV detected in corresponding areas of China were shown in Table S2. We did try to add the information in the map (Figure 2), but it will make it too complicated and hard to read. Therefore, we showed the detailed information in table S2.

In regard to the PPV genome analyses it would be interesting to know whether the variable regions are clustering toward a distinct place within the genome.

Reply: Thank you for this suggestion. As shown in the new added Figure S5, the overall genome homology within each genotype of PPV strains (based on our nearly complete genomes and representative genomes) is from 93.38% to 99.61%. The distributions of mutations are widely spread throughout the genomes but not only be clustering within a specific region of the genome.

Although of minor importance it would make sense to color-code the PPV genotypes in Fig. 3 the same as in Fig. 1.

Reply: Thank your for this suggestion. The color-code of PPV genotype in Figure 3 has been revised same as in Figure 1.

Figure 5 it would be worth indicating changes towards NADL2 (non-virulent) to Kresse (virulent) strains as discussed Lanes 199ff. Are the changes towards the one or the other in agreement with the described pathology? Maybe it would be worthwhile to include such information into this figure.

Reply: Thank you for your suggestions. The distinct residues among NADL-2 vaccine strain (non-virulent), Kresse strain (virulent), our PPV1 viruses and other representative PPV1 sequences were shown in the revised Figure 5. A paragraph of discussion has been added in lines 297-313 to discuss the potential influence of the residues.

Considering the diversity of the different genotypes regarding their genera/subfamily of Parvoviridae it is of interest to comment on the serotypes and potentially cross-neutralizations vs. neutralization escape. If there are valid information in this regard it would be of interest to include this in the discussion.

Reply: Thank you for your comments. There is very limited reports regarding the cross-neutralizations or neutralization escape among different PPVs. A discussion about this issue was added in lines 303-313.

The numbering of the tables does not really follow the flow of the manuscript. Thus Table 1 is part of Mat/Met, Table 3 is preceding Table 2 in paragraph 2 of the results section. This make it difficult to follow.

Reply: Thank you for your suggestions. The numbers of tables and figures have been revised following the flow of the manuscript.

Reviewer #2 (Comments for the Author):

A study of the porcine parvoviral DNA in swine in different regions of China, using PCR to detect the PPV 1-7 sequences. For each virus type some were sequenced, and the sequences analyzed by standard methods (phylogenetics, recombination analysis, and sequence selection). This is very standard in the field and the data is similar to those that have been reported previously in China and other regions of the world, although with more details and some additional sequences. There are no specific surprises here - it has long been known that multiple virus strains co-circulate in swine (and most other animals), and that there is recombination between parvoviruses.

Reply: Thank you for your comments. The manuscript has been revised according to your suggestions as follows.

There are some issues that should be addressed:

1) It should be made clear that the detection of DNA in the samples is not necessarily related to a current active co-infection, but the DNA is likely to be left over from a previous infection of that animal.

Reply: The authors completely agree with the reviewer that the detection of DNA is not identical to a active infection, which has been made clear in the revised manuscript lines 162-167.

2) The recombination is not of the exact viruses listed, but of viruses with sequences related to those viruses.

Reply: Thank you for this critical comment. Yes, the recombination is likely not occurred exactly with the listed viruses. So we revised them as JX15-like isolate and JX38-like isolate throughout the manuscript.

3) There is a lot of repetition between the Results and Discussion - I suggest that the Discussion by rewritten to highlight the main new discoveries, but not of over all of the details again. Likewise, the summary and conclusions largely repeat the Abstract, and can be mostly removed.

Reply: Thank you for your comments. The repetition parts throughout the manuscript have been deleted in the revised version.

October 29, 2021

Dr. Nanhua Chen
Yangzhou University
Preventive Veterinary Medicine
48 East Wenhui Road
R404 International Cooperation Building
Yangzhou, Jiangsu 225009
China

Re: Spectrum01294-21R1 (A systematic investigation unveils high co-infection status of porcine parvovirus types 1 through 7 in China from 2016 to 2020)

Dear Dr. Nanhua Chen:

Your manuscript has been accepted, and I am forwarding it to the ASM Journals Department for publication. You will be notified when your proofs are ready to be viewed.

Sincerely,

Clinton Jones
Editor, Microbiology Spectrum

Journals Department
Supplemental Dataset: Accept
Supplemental Table S1-S4 and Figures S1-S11: Accept